# Cranial Base Synchondrosis: Chondrocytes at the Hub

**DOI:** 10.3390/ijms23147817

**Published:** 2022-07-15

**Authors:** Shawn A. Hallett, Wanida Ono, Renny T. Franceschi, Noriaki Ono

**Affiliations:** 1Department of Periodontics and Oral Medicine, University of Michigan School of Dentistry, Ann Arbor, MI 48109, USA; shallett@umich.edu (S.A.H.); rennyf@umich.edu (R.T.F.); 2Department of Orthodontics, University of Texas Health Science Center at Houston School of Dentistry, Houston, TX 77054, USA; wanida.ono@uth.tmc.edu; 3Department of Diagnostic and Biomedical Sciences, University of Texas Health Science Center at Houston School of Dentistry, Houston, TX 77054, USA

**Keywords:** cranial base, synchondrosis, chondrocyte(s), osteoblast, endochondral ossification, craniofacial development, regeneration, skeleton

## Abstract

The cranial base is formed by endochondral ossification and functions as a driver of anteroposterior cranial elongation and overall craniofacial growth. The cranial base contains the synchondroses that are composed of opposite-facing layers of resting, proliferating and hypertrophic chondrocytes with unique developmental origins, both in the neural crest and mesoderm. In humans, premature ossification of the synchondroses causes midfacial hypoplasia, which commonly presents in patients with syndromic craniosynostoses and skeletal Class III malocclusion. Major signaling pathways and transcription factors that regulate the long bone growth plate—PTHrP–Ihh, FGF, Wnt, BMP signaling and Runx2—are also involved in the cranial base synchondrosis. Here, we provide an updated overview of the cranial base synchondrosis and the cell population within, as well as its molecular regulation, and further discuss future research opportunities to understand the unique function of this craniofacial skeletal structure.

## 1. Introduction

The skull is an important bone structure for human survival—it protects the central nervous system, forms part of the respiratory airway, and enables mastication. Craniofacial skeletal tissues develop, grow, and maintain their functionality throughout life through a series of highly coordinated molecular processes that regulate relevant cell populations. Of the craniofacial bone structures, the cranial base serves as an important component of the skull, containing the cartilaginous synchondroses with autonomous growth potential development. What was previously thought to serve as simply a structural platform for the central nervous system, the cranial base is now regarded as an important autonomous growth center of the craniofacial complex [1]. Unlike the extensively characterized growth plate in long bones [2], little is known about the molecular regulation of the cranial base and its cartilaginous growth center synchondroses. Here, we provide an overview of the unique morphology of the cranial base synchondrosis and the cell population within, as well as its molecular regulation, and discuss future research opportunities to understand the unique function of this craniofacial skeletal structure.

## 2. Overview of the Cranial Base Morphology

### 2.1. Developmental Origins of the Chondrocranium

The development origins of the chondrocranium—the cranial base and its cartilaginous synchondroses—are distinct from those of other endochondral bones in the axial and appendicular skeleton. In mice, neural crest-derived cells migrate ventrally under the forebrain and populate the presumptive cranial base rostral to Rathke’s pouch at E10.5 [3]. These neural crest cells contribute to the most anterior portion of the cranial base, while the more posterior portion of the cranial base is derived from the mesoderm, with the exception of hypochiasmatic cartilages in the most anterior position of the cranial base that are derived from the cephalic paraxial mesoderm [4]. Importantly, paraxial mesoderm remains the principal contributor to both the neurocranium and chondrocranium. Somitomeres, which differentiate from paraxial mesoderm during early embryonic development, undergo divergent morphogenesis to contribute to approximately 80% of the head mesenchyme in mice [5]. These structures, representing the cellular precursors to somites, are critically important for forming the segmental pattern of the cranial mesoderm of later stage embryos. Somitomeres in the cranial region undergo morphogenesis in unison with neural plate development and neuromeric morphogenesis, eventually differentiating into somites [6]. Aberrant somitomere formation may lead to craniofacial skeletal abnormalities.

### 2.2. Development and Growth of the Cranial Base and Its Synchondroses

The neurocranium of the skull contains structures enclosing the central nervous system and its supporting vasculature. These structures include the cranial base (formed by endochondral ossification) and the cranial vault (formed by intramembranous ossification). In mice, the cranial base is composed of several segmented bones including ethmoid, pre-sphenoid, basisphenoid and basioccipital bones. Like the long bones, the cranial base first appears as a sheet of undifferentiated mesenchymal cells in the form of a cartilaginous anlage in early skeletal development. Over time, these cells undergo chondrogenesis to form the chondrocranium, which is a solid cartilaginous structure spanning the entire length of the skull formed by the fusion of several individual cartilages that appear at distinct locations. The cartilages of the chondrocranium then undergo endochondral ossification to form the bones of the cranial base [3]. In humans, the chondrocranium is formed between the 6th–8th week of gestation [7]. The chondrocranium rapidly grows between the 11th–23rd week of gestation, elongating the anterior cranial base antero-posteriorly and enlarging the posterior cranial base transversely [8]. Importantly, in primates, the cranial base flexion directly correlates with brain growth [9]. Thus, there is an intimate relationship between the central nervous system and the chondrocranium.

Separating the bones of the cranial base are the synchondroses, cartilaginous tissues that drive anterior–posterior growth of the skull. Positioned between the pre-sphenoid and basisphenoid bones is the inter-sphenoid synchondrosis (ISS), whereas the more posterior positioned spheno-occipital synchondrosis (SOS) lies between the basisphenoid and basioccipital bones. Similar to the long bone growth plate, the synchondroses are composed of distinct layers of round, proliferating and hypertrophic chondrocytes [10]. However, unlike the unidirectional organization of chondrocytes observed in long bones [11], the synchondroses are bidirectionally organized in a mirror image manner sharing a central round layer (Figure 1). The anisotropic growth mechanism used by these chondrocytes to facilitate anteroposterior growth is not currently understood [12]. The cranial base synchondrosis is composed of a combination of hyaline and fibrous cartilages [13]. During skeletal maturation, chondrocytes proliferate, differentiate into pre-hypertrophic chondrocytes and are eventually replaced by bones at the chondro-osseous junction. Although not examined in the synchondroses, there is evidence in growth plates that chondrocytes can undergo ‘trans-differentiation’ to form osteoblasts [14].

Significantly, in humans, the ISS ossifies at 2–3 years of age, whereas the SOS ossifies much later between 16–18 years in humans [15,16]. While 60% of cranial base growth occurs embryonically, 40% of additional growth occurs postnatally, extending well into adolescence [17]. This emphasizes the importance of the SOS as a potential target for pharmacological intervention in patients with prematurely ossified synchondroses. The anterior positioned ISS is neural crest-derived, while the SOS is comprised of both neural crest cells and paraxial mesoderm in mice [3]. These differences in embryonic origins of the ISS and SOS may contribute to their differential ossification rates and roles in facilitating the elongation of the cranial base. Unlike humans, the synchondroses of mice remain patent throughout adulthood. The basis for these interspecies differences has not been fully elucidated.

### 2.3. Craniofacial Anomalies Associated with Cranial Base Malformation 

Growth of the postnatal cranial base guides overall anteroposterior elongation of the craniofacial complex. Defective cranial base growth in humans leads to midfacial hypoplasia [18], skeletal Class III malocclusion [19], and various forms of syndromic craniosynostoses [20,21]. These patients have significant challenges associated with breathing, speaking, and chewing and often feel socially stigmatized. Invasive surgical intervention, such as Le Fort III osteotomy, is currently the primary treatment option to correct these structural deficiencies.

Malformations of the cranial base are commonly found in craniofacial and skeletal genetic anomalies, including cleidocranial dysplasia and achondroplasia. In cleidocranial dysplasia, which is caused by heterozygous loss-of-function mutations in *RUNX2*, patients present with delayed ossification of the cranial base, leading to midfacial hypoplasia and ‘dish-face like’ appearance, in addition to presenting generalized skeletal malformations [22]. Achondroplasia is associated with premature fusion of the long bone growth plates in addition to premature fusion of the SOS, leading to shortening of the posterior cranial base and midfacial hypoplasia [23]. Additionally, patients presenting with skeletal Class III malocclusion often have significant midfacial hypoplasia, due to precocious ossification of the SOS [24]. In more recent years, several studies have established a relationship between premature ossification rates of the ISS and SOS and various forms of syndromic craniofacial deformities, such as Apert, Pfeiffer and Crouzon [20,21,25], which often present with craniosynostosis. In these studies, closure rates of the synchondroses were assessed radiographically by measuring of ‘open’, ‘partial’ and ‘closed’ synchondroses. The authors determined that the ossification rates of the synchondroses in affected patients were accelerated compared to healthy individuals. These results are potentially important, as there is still debate in the field as to whether the premature fusion of the cranial suture is secondary to the fusion of the synchondrosis fusion or vice versa [26]. Lastly, assessment of cranial base synchondrosis closure rates in patients with Down and Klinefelter syndromes also show premature fusion of the synchondroses, resulting in shortened cranial base length and midfacial hypoplasia [27,28,29]. Collectively, the activity and differentiation trajectories of cranial base chondrocytes, coupled with the fusion rates of the synchondroses, are critical factors in determining anteroposterior elongation of the skull, both in humans and other vertebrate species. Importantly, coupled with the fusion rates of the cranial vault sutures, the ossification rates of the cranial base synchondroses represent unique physiological phenomena that are required for proper organization of the entire craniofacial complex.

## 3. Molecular Regulation of the Cranial Base

To date, limited numbers of studies have directly assessed the molecular regulation of cranial base development. This may be due in part to an underlying assumption that the synchondroses are structurally similar to the long bone growth plate, therefore controlled by the same developmental signals regulating the endochondral pathway. However, the synchondrosis differs from the growth plate in several key aspects. First, unlike long bone growth plates that undergo unidirectional growth, cranial base synchondroses grow in a bidirectional manner with a mirror-image arrangement of the proliferating layers. Second, synchondroses lack an overlying articular synovial layer, which is critical for long bones to withstand mechanical loads. The cranial base is not subjected to direct mechanical loading, due to its proximity to the central nervous system. Third, the cranial base lacks a secondary ossification center (SOC). This may be a critical difference governing behavior, as development of the SOC has been associated with formation of a resident stem cell niche within the resting zone of the postnatal epiphysis [30,31,32]. Presence of a similar microenvironment has not yet been detected in the cranial base. Thus, growth plates and synchondroses may not necessarily share the same molecular regulations. In this review we will compare what is known about the regulation of these two structures with the goal of highlighting areas for future research.

The following molecular regulators are discussed in this review to highlight their contribution to cranial base development: parathyroid hormone–related protein (PTHrP), Indian hedgehog (Ihh) and EvC ciliary complex subunit 1/2 (Evc1/2, encoding Limbin), fibroblast growth factor receptors 1/2/3 (FGFR1/2/3), discoidin domain receptors 1/2 (DDR1/2), Wnt/β-catenin and runt-related transcription factor 2 (Runx2) (Figure 2). This article will complement previous reviews on cranial base development [10,33,34,35].

### 3.1. Role of Parathyroid Hormone Related Protein in Synchondrosis Organization

PTHrP was initially discovered and cloned in 1987 as a factor in the serum of patients with humoral hypercalcemia of malignancy, a skeletal disease associated with bone metastatic cancers [36]. In long bones, a PTHrP-Ihh negative feedback loops are essential for maintaining the mitotic capability of proliferating chondrocytes and initiating the differentiation of these cells into pre-hypertrophic zone chondrocytes [11,37,38]. *Pthrp* is also expressed in resting and proliferating zone chondrocytes in mouse synchondroses [39]. Further, it may have direct functions in this tissue since PTHrP-deficient mice display cranial base abnormalities [40,41]. Specifically, in PTHrP-deficient newborns, chondrocytes in the ISS undergo premature hypertrophy in all layers. Hypertrophy in the SOS of these mice is even more extensive with loss of resting and proliferative layers and only a small hypertrophic chondrocyte layer remaining within the central part of the SOS, surrounded by thick layers of bone-like matrix on both sides. Thus, PTHrP-deficient mice have defective synchondroses, with the SOS undergoing more rapid precocious ossification. This accelerated hypertrophy/ossification of synchondroses is similar to what is seen in long bone growth plates of *Pthrp*-deficient mice [41]. Together, these studies suggest that PTHrP may have similar functions in the synchondroses and long bone growth plate, although further studies will be required to confirm this.

Recently, a subset of resting zone chondrocytes expressing *Pthrp* were identified in long bone growth plates where they function as skeletal stem cells [31], giving rise to columnar chondrocytes in the proliferating zone, hypertrophic chondrocytes as well as osteoblasts and bone marrow stromal cells in the metaphysis. These cells acquire their ‘stem-like’ qualities following the formation of the SOC. Thus, the SOC may have the ability to instruct cells within the resting zone to acquire self-renewal and multi-lineage differentiation potential. Importantly, the cranial base lacks a SOC and, thus, may not possess similar stem cell-inductive capacity. However, the exact relationship between the SOC and this growth plate stem cell population remains to be elucidated. In our recent study, we examined the presence of a *Pthrp*-expressing stem cell population in synchondroses [42]. Surprisingly, *Pthrp* expressing cells were found predominantly within a wedge-shaped structure on the lateral borders of the SOS. PTHrP^+^ cells were not detected in the central resting zone chondrocyte layers of the SOS at any of the early postnatal time points examined. This is in marked contrast to the long bone growth plate where *Pthrp*-expressing cells populate the entire central resting zone. Moreover, lineage-tracing experiments, using a tamoxifen-inducible *Pthrp-creER* line combined with a tdTomato reporter allele, revealed that PTHrP^+^ chondrocytes sparsely occupied the SOS and their progeny failed to establish columnar chondrocytes, whereas in the femur growth plate, these cells robustly labeled the resting zone and differentiated into proliferating and hypertrophic chondrocytes. These recent findings highlight more passive roles of PTHrP^+^ chondrocytes in the postnatal cranial base synchondroses. Importantly, PTHrP^+^ chondrocytes in the cranial base synchondrosis lack the innate capacity to acquire stem cell properties, potentially due to the absence of instructive signals from the SOC.

### 3.2. Indian Hedgehog Is a Critical Regulator of Cranial Base Development

The Hedgehog family of proteins contains three separate members: sonic hedgehog (Shh), desert hedgehog (Dhh) and Indian hedgehog (Ihh). In the long bone growth plate, *Ihh* is expressed by pre-hypertrophic chondrocytes where it facilitates the onset of chondrocyte hypertrophy. Concurrently, Ihh promotes *Pthrp * expression from chondrocytes in the round cell layer, which in turn inhibits *Ihh* expression from pre-hypertrophic chondrocytes, establishing a negative feedback loop with PTHrP [38]. Ihh also induces osteoblast differentiation from progenitor cells in the adjacent perichondrium in conjunction with BMPs [43]. 

The synchondroses of Ihh-deficient mice display significant abnormalities in early skeletal development. More specifically, the SOS and ISS are disorganized and chondrocyte proliferation is markedly suppressed. *Pthrp* expression is downregulated, and chondrocyte maturation is initially delayed, but is subsequently accelerated, as exemplified by the presence of type X collagen-positive chondrocytes within the SOS [39,44]. This suggests that Ihh may directly or indirectly dictate the topographical distribution of chondrocytes in the synchondroses and growth plate. Consequently, the overall anteroposterior elongation of Ihh-deficient mice is markedly decreased compared to wild type littermates. Interestingly, in one study [45], the authors highlight the importance of the overlying perichondrium in establishing the polarity of adjacent chondrocyte zones through the sequestration of extracellular matrix genes [45,46,47]. The authors also postulate that loss of Ihh leads to aberrant function of primary cilia. This is of particular importance, as primary cilia have been shown to regulate chondrocyte polarity [48] and are the sites of Ihh signaling [49].

### 3.3. Primary Cilia EVC/EVC2 as a Regulator of Cranial Base through Hedgehog Modulation

There is now considerable evidence that primary cilia have important functions in organizing the cranial base synchondroses. *Evc/Evc2* (encoding Limbin protein) are N-terminal anchored transmembrane proteins that exist as a complex within the primary cilia where they bind *Smoothened*, a key Hedgehog signaling effector protein [50,51]. Pathogenic mutations in *Evc/Evc2 (Limbin)* in humans cause Ellis-van Creveld syndrome, a disorder associated with a wide range of craniofacial abnormalities, including an enlarged skull, skeletal Class III skeletal malocclusion, skeletal open bite and midfacial hypoplasia [52]. *Evc2* expression is found in nearly all layers of the synchondroses [53]. Further, global knockout of *Evc2*, as well as neural crest cell-specific conditional knockouts (*P0-cre; Evc2^fl/fl^* and *Wnt1-cre; Evc2^fl/fl^*) in mice display midface hypoplasia, resulting from the anterior shortening of the cranial base due to premature fusion of the ISS [54,55]. Moreover, cephalometric analysis of *Evc2*-deficient mice shows significant reductions in overall cranial base length as well as lengths of all segmented cranial base bones [55,56]. These phenotypic observations were also observed in a patient with Ellis-van Creveld syndrome, appearing as pronounced Class III malocclusion [57]. Thus, the primary cilia proteins, EVC/EVC2, play prominent roles in dictating proper cranial base growth and anteroposterior elongation of the midface in both humans and mice.

In a more recent study, the cranial base phenotype of Evc2-deficient mice was described in greater detail [58]. Two key findings were reported: first, chondrocytes in the ISS all became hypertrophic as early as E18.5 in Evc2-deficient mice, whereas a majority of the chondrocytes in the SOS remained undifferentiated. Second, as a consequence of premature ISS hypertrophy, the anterior cranial base was more severely shortened in *Evc2*-deficient mice after E18.5, while the length of the posterior part of the skull floor (controlled by the SOS) was less affected. Thus, targeted *Evc2* deletion more profoundly affects cells of neural crest origin [3]. Given the function of Evc2 in Hedgehog signaling, the authors postulate that premature fusion of the ISS is due to a decreased gradient of *Ihh* expression, which preferentially affects the anterior portion of the ISS relative to the posterior ISS and SOS. Further, resting chondrocytes in *Evc2*-deficient mice undergo premature differentiation into pre-hypertrophic chondrocytes, leading to precocious ossification. The authors hypothesize that during embryonic development, chondrocytes located at the most posterior end of the cranial base cartilage primordia begin differentiating to form the hypertrophic zone, due to an activity of the future basi-occipital bone. Hypertrophic chondrocytes in this zone induce differentiation of chondrocytes at the middle of the cranial base cartilage primordia to form the second hypertrophic zone, which is the future basi-sphenoid bone, and, finally, hypertrophy of the chondrocyte zone for the future pre-sphenoid bone occurs at the anterior end of the braincase floor cartilage primordia. These hypertrophic zones mineralize from posterior to anterior as embryogenesis progresses, and the remaining cartilage will form the intervening ISS and SOS. *Evc2*-deficiency therefore causes aberrant formation of the SOS hypertrophic zone, which is subsequently amplified in the ISS, leading to reduced *Ihh* expression and precocious ossification. Future analyses should seek to examine the molecular regulation controlling the differential ossification rates of the ISS versus SOS, particularly in the context of the regulation of Ihh activation through Evc2 signaling. This is particularly important, as precocious ossification of the anterior portion of the cranial base manifests in several syndromic craniosynostoses, such as Apert, Pfeiffer and Crouzon.

### 3.4. Fibroblast Growth Factor Receptor 3 Regulates Proliferation of Synchondrosis Chondrocytes

Evc2 and Ihh signaling functions to regulate the growth and fusion of the SOS and ISS; however, there are additional pathways that mediate this process. For example, in long bones, FGFR3 is recognized as a target of Ihh signaling that negatively regulates Ihh signaling and inhibits chondrocyte hypertrophy [59]. Therefore, Ihh-Fgfr3 signaling is critical in the context of growth plate development. Yet, whether a similar relationship exists in the synchondroses is currently unknown. 

The mammalian fibroblast growth factor (FGF) family contains eighteen secreted proteins that interact with four tyrosine kinase FGF receptors (FGFRs), performing important functions in embryonic and postnatal development by maintaining progenitor cells and their proliferation [60]. During early skeletal development, FGFs regulate chondrogenesis, osteogenesis, and bone mineral homeostasis [61]. Of the FGF ligands, FGF18 regulates cell proliferation and differentiation during chondrogenesis and osteogenesis [62]. *Fgf18*-deficient mice display shortened long bones and reductions in anteroposterior elongation of the skull, associated with reduced mineralization of possibly aberrant postnatal growth of the cranial base synchondroses. Several other FGF ligands also have significant roles in skeleto-genesis. Ectopic expression of *Fgf2* causes enlarged occipital bones, skeletal dwarfism and coronal suture synostosis in mice [63]. Conversely, Fgf2-deficient mice display premature closure of the growth plate [64]. Targeted overexpression of *Fgf9* in cartilage causes achondroplasia-like phenotypes in mice, characterized by premature closure of the growth plate, associated with significant reduction in the anteroposterior dimension of the skull [65]. Importantly, FGF9 has high affinity for its receptor, FGFR3, and, as discussed later in this review, over-activating mutations in FGFR3 in humans and mice cause achondroplasia and midfacial hypoplasia. Lastly, overexpression of *Fgf3/4* in mice causes a Crouzon syndrome-like phenotype, characterized by premature fusion of the cranial vault sutures and reduced anteroposterior dimension of the skull [66]. The precise roles of the FGF ligands in regulating the cranial base synchondrosis need to be investigated in future studies.

Additionally, in the growth plate, FGFRs 1/2/3 play critical roles in maintaining chondrocyte proliferation and differentiation. For example, FGFR3 signaling diverges to regulate either chondrocyte proliferation via STAT1-p21 [67,68] or hypertrophy via MEK-MAPK pathways [69]. Both signaling cascades are mediated in part by Snail1, which is induced by FGFR3 and plays a role in the manifestation of FGFR-related skeletal dysmorphisms [70]. It is not known if similar signaling cascades exist in the cranial base. Nevertheless, disruption of FGF signaling in a variety of genetic disorders has profound effects on craniofacial development, as is described below.

Of the FGF receptors, FGFR3 mutations have the most profound impact on the skeleton. An activating point mutation in FGFR3 (FGFR3^G308R^) is associated with the most common genetic form of human dwarfism, achondroplasia [71]. Patients present with shortened long bones resulting from premature fusion of the growth plates, due to overactive FGF signaling [72]. In addition, Fgfr3^G308R^ patients show craniofacial phenotypes including a shorter anteroposterior cranial base, due to premature fusion of the synchondroses leading to midfacial hypoplasia^.^ Conversely, conditional deletion of *Fgfr3* in chondrocytes (*Col2a1-cre; Fgfr3^fl/fl^*) causes aberrant upregulation of cell proliferation in the growth plate, leading to the formation of cartilaginous tumors, including osteochondromas and enchondromas [73]. *Col2a1-cre; Fgfr3^fl/fl^* mice displayed slightly elongated skulls. However, the authors did not examine the cranial base in this study. In addition, overactive FGF signaling due to FGFR3^K650E^ gain of function mutations, causes the most common embryonically lethal skeletal dysplasia, thanatophoric dysplasia type II (TDII) [74,75]. TDII fetuses have shortened limbs, straight femurs and cloverleaf-shaped skulls leading to a midfacial hypoplastic phenotype [76]. Although not clinically observed, due to the early lethality of the mutation, these craniofacial manifestations are likely due to congenital malformation of the cranial base synchondroses.

Premature fusion of the synchondroses was shown in a patient with homozygous achondroplasia mutations [77]. In the same study, the authors showed that a mouse model harboring an activating mutation in FGFR3 (e.g., Fgfr3^G374R^) [78], displayed similar premature fusion of the ISS and SOS [77]. The authors showed that constitutive activation of the downstream Fgfr3 effector, MEK1, in chondrocytes (*Col2a1-cre; MEK^S218/222E, Δ32-51^*) [79] caused premature fusion of the synchondroses, recapitulating the phenotype observed in FGFR3 gain-of-function mice. Additional FGFR3 gain-of-function mouse models that display premature ossification of the synchondroses include Fgfr3^369/369^ [80], Fgfr3^P244R^ [81] and Fgfr3^365/+^ [82]. Interestingly, over-activation of FGFR3 in several of these mutants results in decreased expression of *Ihh* in the pre-hypertrophic zone and *Pthrp* in the resting zone of the synchondroses. Disruption of PTHrP-Ihh feedback may be an etiology of the premature ossification phenotype observed in the synchondroses of these Fgfr3 mutant mice.

FGFR1 and 2 also exert significant roles in regulating postnatal cranial base growth and ossification. For example, FGFR1-activating mutations cause Pfeiffer syndrome [83]. This genetic condition is linked to a pericentromic region of chromosome 10 in relation to FGFR1 [84]. Patients with Pfeiffer syndrome commonly present with abnormal fusion of the synchondroses leading to bulging and wide-set eyes, high and prominent forehead and underdeveloped upper jaw leading to midfacial hypoplasia [85]. In a recent study assessing single nucleotide polymorphisms (SNP) in a large patient cohort, two SNPs found in FGFR1 were correlated with overall head size and midfacial development [85]. The authors discovered that modified FGFR1 alleles harboring SNPs had a small midface and hypertelorism, which are hallmark craniofacial features of Pfeiffer Syndrome. Thus, although FGFR1’s role in craniofacial development is largely unknown, more recent observations point to its role in the cranial base synchondrosis.

Apert syndrome is one of the most common syndromic craniosynostoses [86]. Apert syndrome is caused by activating mutations in FGFR 2 (S252W or P253R) and clinically manifests as premature fusion of the cranial vault sutures and cranial base synchondroses, leading to midfacial hypoplasia and frontal bossing [87]. Fgfr2^P253R^ mice display abnormalities in both osteogenesis and chondrogenesis, including premature closure of the coronal suture and delayed growth of the growth plate and cranial base synchondroses leading to shortened midfaces [88]. Blockade of downstream FGFR2 signaling via in vitro inhibition of Erk1/2 with PD98059 partially rescued the calvarial and long bone defects in Fgfr2^P253R^ mice. More specifically, in the growth plate, treatment caused the partial elongation of the proliferating, hypertrophic and calcifying zones. However, the cranial base synchondroses were not evaluated. In an additional study, Fgfr2^P253R^ mice displayed premature fusion of the SOS and ISS due to accelerated differentiation of proliferating and hypertrophic chondrocytes, resulting in midfacial hypoplasia [89]. FGFR2^S252W^ mutations in humans and mice have also been associated with Apert syndrome [90,91]. Similar to the Fgfr2^P253R^ mutant phenotype, patients harboring the FGFR2^S252W^ mutation present several craniofacial phenotypes, including hypertelorism, frontal bossing and midfacial hypoplasia [90]. In Apert patients harboring this mutation, the rate at which the SOS fuses is accelerated by 1–3 years when compared to unaffected individuals [92]. Fgfr2^S252W^ mutant mice present a similar midfacial hypoplasia phenotype as humans, in addition to long bone defects due to delayed formation of the SOC [91]. Although the authors did not assess the cranial base, these animals present with significant shortening of all growth plate zones. Thus, the midfacial hypoplastic phenotype observed in Fgfr2 mutant mice may be the result of aberrant development of the cranial base synchondroses. In an additional study, mesoderm-specific or neural crest-specific over-activation of FGFR2 signaling (*Mesp1-cre; Fgfr2^S252W^* or *Wnt1-cre; Fgfr2^S252W^*) in mice causes midfacial shortening and significant curvature of the cranial base [93]. In summary, FGFR2 over-activation is a significant contributor to the etiology of Apert syndrome. How it directly impacts on the cranial base synchondrosis requires further investigation.

Methods to reverse the pathogenicity of activating mutant FGFRs are in development [94]. Currently, the only treatment modality available for patients is multiple rounds of surgery. Efforts to revert the pathogenicity of overactive FGFR3 using FGFR3 inhibitors, statins, meclozine and C-type natriuretic peptide are also being explored. In a recent study, Fgfr3^G380R^ mice treated with Recifercept, a soluble FGFR3 protein inhibitor that mimics the extracellular domain of FGFR3, and, thus, competes with FGFR3 for ligand binding, resulted in the stabilization of endogenous downstream signaling of the protein [95]. Fgfr3^G30R^ mice treated with Recifercept restored normal body weight and prevented premature ossification of the SOS. In these animals, skull length, length/width skull ratio and foramen magnum area were all improved after 3-week treatment with Recifercept [96]. In another study, statins were found to ameliorate aberrant chondrocyte differentiation in a human induced pluripotent stem (iPS) cell model of thanatophoric dysplasia type I, characterized by overactivated FGFR3 signaling [97]. Rosuvastatin stimulates bone elongation in *Fgfr3^G380R^* mice, associated with increased expression of *Sox9, Col2a1, Acan, Runx2* and *Col10a1*, and a rescue of the achondroplasia phenotype. Furthermore, meclozine, an anti-histamine used to treat motion sickness, inhibits FGFR3 activity in chondrocyte cell lines and in a mouse model of achondroplasia, where it rescued the skeletal defects [98,99,100]. Lastly, C-type natriuretic peptide inhibits aberrantly activated FGFR3-mediated MAPK signaling in achondroplasia, thereby restoring longitudinal growth of endochondral bones. Further, overexpression of *Nppc*, which regulates C-type natriuretic peptide production, ameliorates the skeletal and craniofacial abnormalities associated with the achondroplasia phenotype in mice [101,102]. Thus, approaches aimed at reversing the pathogenic effects of FGFR3 overactivation and other FGFRs appear to be effective.

### 3.5. Discoidin Domain Receptors Play Critical Roles in the Organization and Maintenance of Cranial Base Synchondrosis Chondrocytes

The discoidin domain receptors (DDRs) are a class of receptor tyrosine kinases (RTKs) that, together with β1 integrins, mediate interactions between cells and the extracellular matrix by specifically binding triple helical collagens [103,104]. As described below, disruption of DDR signaling can have profound effects on cranial base function.

DDR1 was identified in a screen for tyrosine kinase proteins expressed in human malignancies, but has more recently been shown to play roles in mediating tumor progression of soft tissue carcinomas [105,106,107]. In long bones, conditional ablation of DDR1 (*Col2a1-cre; Ddr1^fl/fl^*) in mice causes hind limb abnormalities, including delayed formation of the SOC, decreased body length and weight, decreased chondrocyte proliferation and apoptosis and delayed chondrocyte hypertrophy [108]. Although the skulls of these animals were not analyzed, the anteroposterior length of the skull appeared to be reduced, potentially due to aberrant development of the cranial base synchondroses. Thus, although there is only limited evidence available, DDR1 may play an important role in cranial base development; thus, further studies are needed to examine its putative role in the synchondroses.

DDR2 mutations in humans cause spondylo-meta-epiphyseal dysplasia (SMED), classified by shortened limbs, abnormal calcifications and craniofacial phenotypes including prominent forehead, open fontanelles, hypertelorism and midfacial hypoplasia [109,110,111,112]. In mice, *Ddr2* is expressed in a gradient fashion in the long bone growth plate, with highest levels in the resting zone and lowest in the hypertrophic zone [113]. In the cranial base synchondroses, a similar expression gradient is observed [114]. Lineage tracing using a tamoxifen-inducible *Ddr2-creER* line showed that Ddr2-labeled cells are present in the cranial sutures and the resting and proliferating zones of the ISS and SOS, which could then differentiate into hypertrophic chondrocytes, osteoblasts and osteocytes, supporting the concept that *Ddr2* is expressed by skeletal progenitors. Ddr2-deficient mice display craniofacial abnormalities, including significant reductions in the lengths of the anterior and posterior cranial base and reduced mineralization of the frontal suture. Defects in cranial base length were related to reduced proliferation in both ISS and SOS chondrocytes in the absence of altered apoptosis or reduced osteoblast differentiation [114]. Interestingly, the reduced cranial base growth was associated with widening of synchondroses, loss of chondrocyte polarity and abnormal distribution of *ColX* and *Col2*, suggesting a role for Ddr2 in regulating fibrillogenesis and extracellular matrix distribution.

In the same study, the authors observed partial colocalization of Ddr2 with the putative skeletal stem cell marker, Gli1 [115]. Conditional deletion of *Ddr2* in Gli1^+^ skeletal progenitors (*Gli1-creER; Ddr2^fl/fl^*) caused a similar cranial base phenotype to that seen in globally deficient mice, including widening of the ISS and SOS and significant reductions in the lengths of the segmented cranial base bones, leading to midfacial hypoplasia. Additionally, conditional ablation of *Ddr2* in chondrocytes (*Col2a1-creER; Ddr2^fl/fl^*) caused a similar cranial base phenotype as observed in the other two mutants. The authors postulate that reductions in cranial base endochondral growth underlying midfacial hypoplasia are likely due to the following: (1) inhibition of resting-to-proliferating chondrocyte differentiation leading to inhibited chondrocyte proliferation and (2) misalignment of synchondrosis chondrocytes with loss of cell polarity, causing aberrant chondrocyte orientation and thus improper local differentiation gradients. 

In summary, DDR2 functions in skeletal progenitors to control synchondrosis extracellular matrix organization, chondrocyte proliferation and polarity and formation of the cranial base bones. It is unknown if DDR2 has a direct role in regulating activity of major regulatory pathways in the synchondroses, such as PTHrP-Ihh, Fgfr3 or Wnt signaling. Yet, chondrocyte abnormalities and cranial base growth defects presented in *Ddr2*-deficient and conditional knockout mice would suggest a functional role for DDR2 in these pathways. In the future, DDR2 function should be examined in the context of other signaling cascades that are pertinent to cranial base development, in addition to other (non)-syndromic diseases manifesting abnormalities in the cranial base synchondrosis.

### 3.6. Bone Morphogenic Proteins Are Expressed in the Cranial Base Synchondroses

Unlike the DDR proteins, which function to establish and maintain the extracellular matrix of the growth plate and synchondroses, the Bone Morphogenetic Proteins (BMPs) play important roles in regulating the formation of bone and cartilage and overall skeletal development [116]. BMP ligands induce bone formation in physiological, pathological and therapeutic contexts. However, this has primarily been shown in long bone fracture healing and periodontal regeneration [117,118]. In the cranial base, BMP function remains largely uninvestigated. Several studies have shown that BMP receptors and ligands are expressed in the embryonic and postnatal cranial base synchondroses. *Bmp2*, *Bmp5*, and *Bmp6* are expressed in the early mesenchymal condensations [119]. During early postnatal development, *Bmp2*, *Bm3*, *Bmp4*, and *Bmp5* were detected in the perichondrium and in the adjacent mesenchyme. Subsequently, *Bmp2* and *Bmp6* expression was confined to hypertrophic chondrocytes, while *Bmp3*, *Bmp4*, and *Bmp5* were expressed in osteoblasts of the trabecular bone and bone collar. Interestingly, *Bmp3* is expressed in the presumptive central hypertrophic zone of the SOS [119]. These expression data suggest that BMPs may play critical roles in mediating chondrocyte proliferation and differentiation, and possibly, regulating chondrocyte-to-osteoblast transition in the synchondroses. BMPs have been shown to possess similar functions in long bones [116]. As would be expected if BMPs function in the cranial base, the Bmp receptors, *Alk6* and *Bmpr1b* are expressed in the embryonic cranial base anlagen [120,121]. In a more recent study, constitutive activation of Bmpr1a in neural crest cells (*P0-Cre; caBmpr1a*) caused precocious ISS ossification during postnatal development. More specifically, *P0-Cre; caBmpr1a* mice displayed significant reductions in the width of the ISS proliferating zone, decreased cell proliferation and increased apoptosis in the overlying perichondrium [122]. Thus, proper regulation of Bmpr1a levels in neural crest cells is critically important for the formation of the ISS. Given the major roles of BMPs in endochondral bone growth, the functions of the BMP ligands and their receptors should be further examined in the context of cranial base formation, maintenance and ossification.

### 3.7. Canonical and Non-Canonical Wnt/β-Catenin Signaling in Synchondrosis Formation

The Wnt/β-catenin pathway functions, in mesenchymal progenitor cells and chondrocytes, to control endochondral bone osteoblasts [123]. In long bones, canonical Wnt signaling dampens chondrogenic differentiation, but promotes chondrocyte maturation and hypertrophy [124,125,126]. In the cranial base, conditional deletion of β-catenin in undifferentiated mesenchymal cells using *Dermo1-cre* (*Dermo1-cre; Ctnnb1^fl/f/^*) increased synchondrosis cartilage formation and decreased osteoblast formation [127]. Additionally, conditional deletion of β-catenin in chondrocytes (*Col2a1-cre; β-catenin^fl/fl^*) caused aberrant replacement of osteoblasts with chondrocytes during embryonic development, resulting in abnormal retention of chondrocytes [128]. More specifically, mutant mice had less-defined chondrocyte layers associated with ectopic *ColX* and *Mmp13* expression outside the synchondrosis hypertrophic zones. The authors also noted abnormal distribution of *Ihh* and *Pthrp* in mutant synchondroses, suggesting that loss of β-catenin in synchondrosis chondrocytes leads to mis-regulation of PTHrP-Ihh feedback. Additionally, these mice had significant reductions in the anteroposterior elongation of the skull, due to reductions in cranial base bone lengths. In the same study, the authors also showed that constitutive activation of canonical Wnt/β-catenin signaling via constitutive activation of Lef1 (*CA-Lef1*) in chondrocytes caused accelerated differentiation of all chondrocyte layers, increased expression of *Col10a1* and the post-hypertrophic/bone marker, *Opn*, in the synchondrosis. Thus, canonical Wnt/β-catenin signaling is important for regulating chondrocyte differentiation in the cranial base synchondrosis.

In long bones, non-canonical Wnt/planar cell polarity (PCP) signaling plays a key role in facilitating asymmetric divisions of resting chondrocytes [129]. The hallmark characteristics of growth plate chondrocytes are oriented division, rearrangement and intercalation of chondrocyte clones in the resting zone, and their subsequent asymmetric divisions into their daughter cells aligned with the axis of growth [130]. Few studies to date have examined the role of the Wnt/PCP pathway in the cranial base synchondroses. Prickle1 is a critical regulator of Wnt/PCP signaling that interacts intracellularly with PCP proteins, including Dishevelled (Dsh), resulting in the modulation of Wnt/PCP and canonical Wnt/β-catenin signaling pathways [131,132]. *Prickle1*-deficient mice display skeletal and craniofacial abnormalities, including misalignment of growth plate chondrocytes leading to shortened bones and midfacial hypoplasia, respectively [133]. Prickle^Bj/Bj^ mice harboring a single nucleotide non-synonymous mutation (c.G482T:p.C161F) have defects in endochondral ossification, resulting in shortened long bones and significant abnormalities in the cranial base synchondroses [134]. More specifically, Prickle^Bj/Bj^ mice displayed shortened primary ossification centers in the cranial base and absence of well-defined chondrocyte layers, including replacement of proliferating chondrocytes with resting cells. Additionally, mutant synchondroses displayed expanded *Sox9*, *Col2* and *Ihh* expression domains. *ColX* and *Alp* expression patterns extended across the entire hypertrophic zone, versus being restricted to a defined layer of cells in controls. These expression patterns suggest that mutant chondrocytes remain immature, thereby delaying ossification of the synchondroses. The authors postulate that the abnormalities observed in all chondrocyte layers of Prickle^Bj/Bj^ mice are potentially due to impaired binding of Prickle^Bj^ to Dsh2/3, leading to the loss of chondrocyte polarity, random distribution of core PCP proteins in synchondrosis chondrocytes and altering of the plane of cell division of chondrocyte layers. Thus, the Wnt/PCP pathway plays critical roles in the orientation, alignment and differentiation trajectories of synchondrosis chondrocytes through early cranial base development.

### 3.8. Runt Related Transcription Factor 2 Is a Novel Regulator of Cranial Base Development

We have described the functional roles of various signaling cascades in the context of the development of the cranial base synchondrosis. Importantly, Runt-related Transcription Factor 2 (Runx2) has been previously shown to be a putative downstream target of several key regulatory pathways in endochondral bone formation. However, its function in the context of cranial base development remains largely unknown. In the following, we provide an update to our current understanding on the Runx2’s role in the formation, maintenance and eventual fusion of the cranial base synchondroses in an attempt to highlight this promising target.

Runx2 is a master regulator of osteoblast and hypertrophic chondrocyte differentiation in long bones [135,136]. *Runx2*-deficient mice display deficient skeleto-genesis due to aberrant chondrocyte and osteoblast function. *Runx2* expression is also found in chondrocytes of the cranial base synchondroses and osteoblasts in the adjacent region [135]. In humans, RUNX2 haploinsufficiency causes cleidocranial dysplasia (CCD), associated with hypoplastic clavicles, patent sutures and fontanelles, supernumerary teeth and short stature [137]. CCD patients display midfacial hypoplasia, due to persistent cartilage formation in the cranial base synchondroses. However these results are based on radiographic interpretation alone [138]. Constitutive activation of *Runx2* in Col2a1^+^ chondrocytes partially rescued the aberrant skeleto-genesis phenotype caused by global *Runx2* ablation in [139]. Moreover, these mice displayed accelerated hypertrophic chondrocyte differentiation leading to premature ossification of long bone growth plates. 

Current knowledge of Runx2 function in cranial base development is somewhat limited, yet, in a recent study, T-box transcription factor 1 (Tbx1) was discovered as a negative regulator of Runx2 during cranial base development [140]. Pathogenic mutations in TBX1 in humans are associated with DiGorge and velocardiofacial syndromes, which present with microcephaly, low-set ears, and cleft palate [141]. *Tbx1*-deficient mice display precocious ossification of the SOS, but not the ISS, leading to premature fusion of the basisphenoid and basioccipital bones and midfacial hypoplasia [140]. Mesoderm-specific deletion of *Tbx1* (*Mesp1-cre; Tbx1^fl/fl^*) recapitulated this phenotype, suggesting a functional role for Tbx1 in cells of the mesoderm lineage. Analysis of signaling cascades involved in Tbx1-mediated downstream gene regulation, using a proximity ligation assay, provided evidence for the involvement of Runx2. Tbx1 was discovered to directly inhibit Runx2 transcriptional activity through its Runt domain. The authors postulate that *Tbx1* ablation causes the upregulation of Runx2 and, subsequently, the upregulation of its downstream targets, *Sp7*, *Col1a1*, *Vegfa*, and other factors involved in vascular invasion into the hypertrophic chondrocyte and osteogenic layers [142]. This indirect upregulation of osteoblastic genes may be involved in the precocious SOS ossification phenotype observed in *Tbx1*-deficient mice. Additionally, histone deacetylase 4 (HDAC4) has been shown to inhibit Runx2 activity [143]. HDAC4 is activated by PTHrP-promoted dephosphorylation, resulting in the initiation of HDAC4’s translocation into the nucleus [144]. *Hdac4*-deficient mice exhibited premature ossification of the synchondroses in [143]. Thus, PTHrP-HDAC4-Runx2 signaling may represent a critical axis that controls postnatal cranial base development and ossification.

Previous studies have begun to shed light on a novel signaling cascade involving Runx2 that is critical for maintaining postnatal cranial base organization. For example, several studies have described a Fgfr3-Mapk-Runx2 signaling axis that controls osteoblast and chondrocyte differentiation in long bones and calvarial osteoblasts [145,146]. As described earlier, FGFR3-MAPK signaling is required for proper postnatal growth of the cranial base. Further investigation into its overlapping functions with PTHrP-HDAC4 and FGFR3-MAPK signaling pathways may unravel a novel biological paradigm regulating cranial base synchondrosis development. However, whether this relationship exists in the context of cranial base development remains unknown. Collectively, Runx2 may possess both direct and indirect functions in regulating the cranial base synchondrosis. 

## 4. Conclusions

The cranial base, once believed to function simply as a structural platform for the central nervous system, is now recognized to have unique functions in shaping the vertebrate craniofacial complex. Although the cranial base and long bone growth plate have similar endochondral origins and contain comparable cell types, the molecular regulation of these two tissues may be vastly different (Figure 2). Using recent technological advances, such as in vivo lineage-tracing, single-cell and spatial transcriptomics and CRISPR screening assays, more detailed analyses of the cellular and molecular properties of the cranial base synchondroses are now possible [147]. Importantly, while these technologies offer significant benefits, they may also present inherent challenges. For example, use of a transgenic line expressing Aggrecan enhancer-driven, Tetracycline-inducible *cre* that is sufficient to activate the *Col2a1* promoter caused unexpected precocious ossification of the ISS and SOS likely because of *cre*-mediated toxicity [148,149]. Thus, further optimization of these techniques is required to discern unique biological phenomena. 

This article summarized our current understanding of the molecular regulation of cranial base formation. However, additional investigation is required to elucidate the molecular etiology of cranial base malformations, as well as to identify potential targets for clinical intervention. In doing so, we become one step closer to bridging the gap between basic knowledge and clinical applications related to the development of novel regenerative and pharmacological therapies for patients suffering from congenital cranial base malformations and traumatic injuries to the craniofacial skeleton.

## Figures and Tables

**Figure 1 ijms-23-07817-f001:**
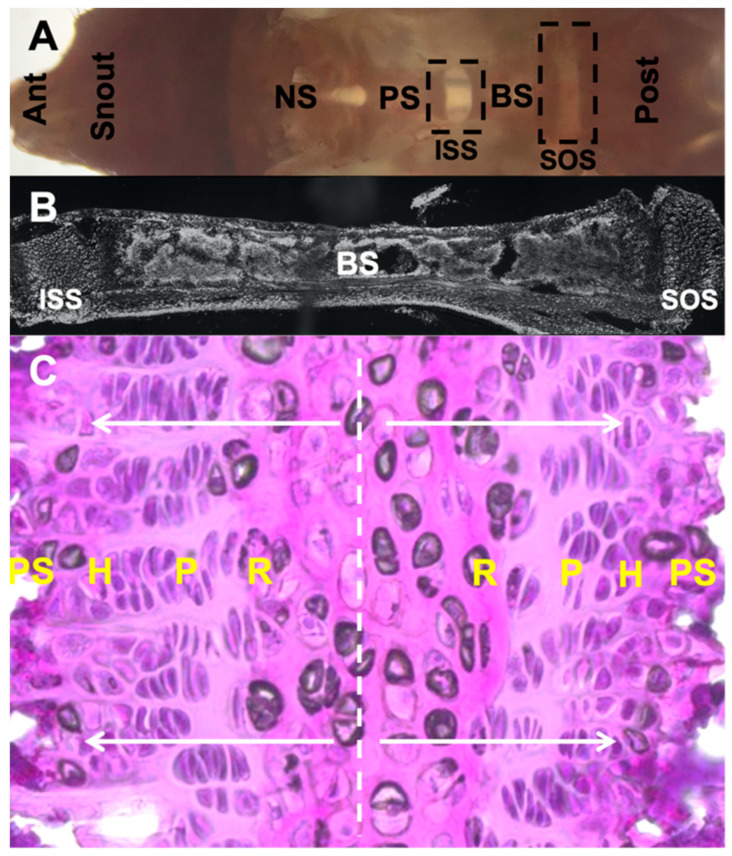
Morphology of cranial base and SOS harvested at postnatal day 28 from C57BL/6 mouse. (**A**). Gross morphology of dissected cranial base positioned in an anteroposterior manner. (**B**). Sagittal section of ISS, basi-sphenoid bone and SOS. (**C**). Magnified image highlighting bidirectional arrangement of chondrocyte layers in SOS stained with hematoxylin and eosin and presence of presumptive central hypertrophic chondrocytes. Ant: anterior, NS: nasal septum, PS: pre-sphenoid, BS: basi-sphenoid, Post: posterior, ISS: inter-sphenoid synchondrosis, SOS: spheno-occipital synchondrosis, R: resting zone, P: proliferating zone, H: hypertrophic zone, PS: primary spongiosa.

**Figure 2 ijms-23-07817-f002:**
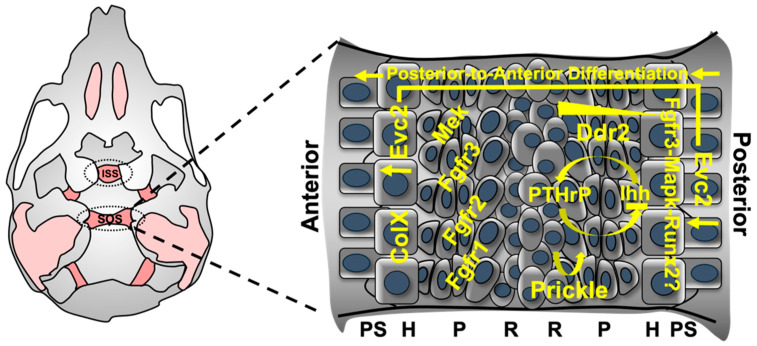
Molecular regulation of the postnatal cranial base synchondrosis. PTHrP-Ihh feedback regulates activity of proliferating chondrocytes until their differentiation into pre-hypertrophic cells. Prickle, a component of the Wnt/PCP pathway, directs asymmetric division of resting cells into adjacent columnar chondrocytes. *Fgfr1*, *Fgfr2* and *Fgfr3* expression in proliferating chondrocytes directly controls cranial base elongation. Activating mutations in each receptor are associated with various forms of syndromic craniosynostoses, which commonly present midfacial hypoplasia. *Evc2* expression is localized in a gradient-like fashion, wherein which levels increase in the anterior most portions of the SOS and ISS are indirectly regulated through Hedgehog signaling. This relationship instructs posterior-to-anterior differentiation and eventual mineralization of cranial base chondrocytes in addition to formation of the segmented cranial base bones. *Ddr2* expression is found in a gradient fashion, with highest levels localized to resting cells. Currently established in the long bone growth plate, Fgfr3-Mapk-Runx2 signaling is a critical component regulating both chondrocyte proliferation and hypertrophy. Whether this relationship exists in the cranial base synchondrosis remains to be elucidated. ISS: inter-sphenoid synchondrosis, SOS: spheno-occipital synchondrosis, R: resting zone, P: proliferating zone, H: hypertrophic zone, PS: primary spongiosa.

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
