# Peer review of "Cranial Base Synchondrosis: Chondrocytes at the Hub"

_ijms, 2022, doi:10.3390/ijms23147817_

Round 1

Reviewer 1 Report

The authors of this narrative review article undertook to elucidate the features of a very special population of chondrocytes and cartilage in the cranial base. In my opinion, this is a very interesting and at the same time quite neglected topic, despite its importance in cranial malformations. Indeed, a much better understanding of these cartilages and chondrocytes within is required. The authors provide a detailed summary of the molecular players regulating cranial base formation, and also pinpoint areas that need further studies. 

What I felt was missing from this otherwise very nicely written paper is a more general introduction to the early embryonic development of the skull, introducing key concepts such as neural crest, paraxial mesoderm, somites and somitomeres, and how they all interact to contribute to the development of the cranial base and the synchondroses within. In its present form, there are implications of these events, but the bigger picture as missing. Therefore, I suggest that the authors add a paragraph somewhere at the beginning of the manuscript describing these events.

The other thing I felt was a bit confusing is that at places the authors describe events happening in the mouse, whereas at other places they describe human events and disorders. I had the feeling that these were a bit messy and confusing. For example, on page 2, paragraph 1 (line 43) the description relates to mice; however, I was missing the human side of the story. I suggest adding some sentences of the human chondrocranium and the synchondroses within.

Just out of curiosity; what type of cartilage is present in synchondroses (hyaline or fibrous)? Are these chondrocytes similar in morphology to those present in articular cartilage? Perhaps these points should also be included in the introduction to give some points of reference to the more general reader.

Minor comments

1. Line 24, the test reads: “forms the respiratory airway” - I think this should be more specific; I would recommend adding “part of the respiratory airway”.

2. In figure 1, SOS is not labelled in the figure (panel B).

3. On page 3, line 95, the authors describe the symptoms of cleidocranial dysplasia, but their description is limited to malformation of the head. Therefore, I would suggest adding a short sentence such as “in addition to generalised skeletal malformations”.

4. Lines 104-105: the text reads: “synchondroses were accelerated”. Please could the authors rephrase the sentence.

5. Line 240: What do the authors mean by “Class III skeletal pattern”? Please elaborate.

Author Response

Major points

  1. The authors of this narrative review article undertook to elucidate the features of a very special population of chondrocytes and cartilage in the cranial base. In my opinion, this is a very interesting and at the same time quite neglected topic, despite its importance in cranial malformations. Indeed, a much better understanding of these cartilages and chondrocytes within is required. The authors provide a detailed summary of the molecular players regulating cranial base formation, and also pinpoint areas that need further studies. 

Thank you very much for your positive assessment on our manuscript.

  1. What I felt was missing from this otherwise very nicely written paper is a more general introduction to the early embryonic development of the skull, introducing key concepts such as neural crest, paraxial mesoderm, somites and somitomeres, and how they all interact to contribute to the development of the cranial base and the synchondroses within. In its present form, there are implications of these events, but the bigger picture as missing. Therefore, I suggest that the authors add a paragraph somewhere at the beginning of the manuscript describing these events.

Thank you very much for your suggestion. Following the reviewer’s suggestion, we have now added a new section (Section 2.1, Developmental origins of the chondrocranium) in the revised manuscript (Line 39-72).

  1. The other thing I felt was a bit confusing is that at places the authors describe events happening in the mouse, whereas at other places they describe human events and disorders. I had the feeling that these were a bit messy and confusing. For example, on page 2, paragraph 1 (line 43) the description relates to mice; however, I was missing the human side of the story. I suggest adding some sentences of the human chondrocranium and the synchondroses within.

Thank you for your suggestion. Following the reviewer’s suggestion, we have now added new sentences at the end of the 1st paragraph in Section 2.2. Development and growth of the cranial base and its synchondroses) (Line 84-89).

  1. Just out of curiosity; what type of cartilage is present in synchondroses (hyaline or fibrous)? Are these chondrocytes similar in morphology to those present in articular cartilage? Perhaps these points should also be included in the introduction to give some points of reference to the more general reader.

We have now clarified this in the revised manuscript (Section 2.2. Development and growth of the cranial base and its synchondroses) (Line 100-101).

Minor comments

  1. Line 24, the test reads: “forms the respiratory airway” - I think this should be more specific; I would recommend adding “part of the respiratory airway”.

We have updated the text as requested (Line 24).

  1. In figure 1, SOS is not labelled in the figure (panel B).

We have added the caption as suggested.

  1. On page 3, line 95, the authors describe the symptoms of cleidocranial dysplasia, but their description is limited to malformation of the head. Therefore, I would suggest adding a short sentence such as “in addition to generalised skeletal malformations”.

We have updated the text as suggested (Line 309-310).

  1. Lines 104-105: the text reads: “synchondroses were accelerated”. Please could the authors rephrase the sentence.

We have rephrased the sentence as requested (adding “the ossification of”, Line 319).

  1. Line 240: What do the authors mean by “Class III skeletal pattern”? Please elaborate.

We have updated the text to read “Class III skeletal malocclusion” (Line 449).

Reviewer 2 Report

To this reviewer' s knowledge, few comprehensive review articles have been published regarding synchondrosis in comparison with the growth plate cartilage.  Therefore, such a unique review may be considered for publication after the authors duly address several issues itemized below.   

Major points

1) The contents of present Abstract and Introduction are quite alike.  The authors are expected to present a summary rather than introduction in the Abstract.  Therefore, reconsideration of the present Abstract is recommended.

2) The image in Figure 1C, which is supposed to be a differential interference microscopic image, is not sufficiently clear.  If available, another image of SOS stained by a regular method (Alcian blue-PAS, for example) may be feasible to clearly show the distinct layers of chondrocytes.

3) In the PDF file provided, only one panel can be found for Figure 2, whereas legends to panels B and C, which are similar to those for Figure 1, are included.  Please check this and make necessary corrections. 

4) For the better understanding of readers, an additional figure with an illustration that explain the authors' hypothesis explained in "3.3. Primary Cilia EVC-EVC2 as a Regulator of Cranial Base through Hedgehog Modulation" may be considered. 

5) In "3.4. Fibroblast Growth Factor (Receptor) 3 Regulates Proliferation of Synchondrosis Chondrocytes", the authors ought to summarize the roles of, not only FGF receptors, but also FGF ligands in synchondrosis. 

6) Recent studies indicated the effectiveness of statins, meclozine and C-type natriuretic peptide on achondroplasia.  These topics need to be included as well. 

Minor point

3.4. Fibroblast Growth Factor 3 Regulates Proliferation of Synchondrosis Chondrocytes: As already pointed out above, the authors probably meant Fibroblast Growth Factor Receptor 3 rather than Fibroblast Growth Factor 3 in the subsection title.  Similarly, the first sentence of the fifth paragraph may read "FGFR1 and 2" instead of "FGF1 and 2".

Author Response

(The authors gave the same response as above.)

Round 2

Reviewer 2 Report

According to the revised manuscript, most of the issues pointed out by this reviewer have been duly addressed; however, the authors' response given is not to the comments of this reviewer, but to those of the other reviewer.  This mistake could happen either at author, or editor level.  I just would like to read the real authors' response to my comments before suggesting acceptance.